# Differences and Changes in Cerebellar Functional Connectivity of Parkinson’s Patients with Visual Hallucinations

**DOI:** 10.3390/brainsci13101458

**Published:** 2023-10-13

**Authors:** Liangcheng Qu, Chuan Liu, Yiting Cao, Jingping Shi, Kuiying Yin, Weiguo Liu

**Affiliations:** 1Link Sense Laboratory, Nanjing Research Institute of Electronic Technology, Nanjing 210019, China; 18629564627@163.com (L.Q.); liuchuan_1007@163.com (C.L.); 2Department of Geriatrics, Affiliated Brain Hospital of Nanjing Medical University, Nanjing 210029, China; 13218000352@163.com (Y.C.); profshijp@163.com (J.S.)

**Keywords:** Parkinson’s disease, visual hallucinations, cerebellum, functional connectivity

## Abstract

Recent studies have discovered that functional connections are impaired in patients with Parkinson’s disease (PD) accompanied by hallucinations (PD-H), even at the preclinical stage. The cerebellum has been implicated in playing a role in cognitive processes. However, the functional connectivity (FC) between the cognitive sub-regions of the cerebellum in PD patients with hallucinations needs further clarification. Resting-state functional magnetic resonance imaging (rs-fMRI) data were collected from three groups (17 PD-H patients, 13 patients with Parkinson’s disease not accompanied by hallucinations (PD-NH), and 26 healthy controls (HC)). The data were collected in this study to investigate the impact of cerebellar FC changes on cognitive performance. Additionally, we define cerebellar FC as a training feature for classifying all subjects using Support Vector Machines (SVMs). We found that in the PD-H patients, there was an increase in FC within the left side of the precuneus (PCUN) compared to the HC. Additionally, there was an increase in FC within the bilateral opercular part of the inferior frontal gyrus (IFGoprec) and triangular part of the inferior frontal gyrus (IFCtriang), as well as the left side of the postcentral gyrus (PoCG), inferior parietal lobe (IPL), and PCUN compared to the PD-NH patients. In the machine learning training results, cerebellar FC has also been proven to be an effective biomarker feature, achieving a recognition rate of over 90% for PD-H. These findings indicate that the cortico-cerebellar FC in PD-H and PD-NH patients was significantly disrupted, with different patterns of distribution. The proposed pipeline offers a promising, low-cost alternative for diagnosing preclinical PD-H and may also be beneficial for other degenerative brain disorders.

## 1. Introduction

PD is a common and complex degenerative disease of the central nervous system in middle-aged and older people. The main pathological change in this condition is the loss of dopaminergic neurons in the substantia nigra pars compacta of the midbrain. Clinically, it mainly presents with motor symptoms such as bradykinesia, resting tremor, and postural instability, as well as various non-motor symptoms [1]. The non-motor symptoms include autonomic dysfunction, psychiatric disorders, sleep disorders, and sensory symptoms [2]. Hallucinations are a common psychiatric symptom, which can manifest as visual, auditory, or olfactory hallucinations. Among these, visual hallucinations are the most prevalent, affecting over 40% of patients with PD. As the disease advances, this percentage increases to 80% [3]. Visual hallucinations severely reduce the quality of life for patients with Parkinson’s disease, increase hospitalization rates, burden nursing staff, and are somewhat associated with mortality rates. [4,5].

Many patients with PD experience sleep disorders characterized by rapid eye movement abnormalities, which can manifest as vivid dreams or minor hallucinations. Whether or not the patient is experiencing hallucinations requires a thorough examination of their medical history and an understanding of the specific details of the hallucinations. A broad scale is currently available, but the scores on the scale may be biased due to various factors. Many patients with neurodegenerative diseases experience visual hallucinations, including Lewy body dementia, cerebrovascular diseases, and frontotemporal dementia. However, only individuals with PD still maintain good cognitive abilities and can communicate effectively with others. This is beneficial because it helps us understand their hallucinations and provide assistance for our experiments.

Resting-state functional magnetic resonance imaging (rs-fMRI) is an in vivo functional imaging technique that measures blood-oxygen-level-dependent (BOLD) signals while scanning subjects at rest, without any explicit task involvement. It is widely used in the diagnosis and prediction of the disease progression of PD [6]. The application of resting-state fMRI techniques has revealed imaging features of PD related to brain structure [7]. Studies have shown that FCs in PD are impaired even in the early stages of hallucinatory concomitants [8]. Holroyd found higher activation in the frontal and subcortical lobes, as well as lower activation in the visual cortex, during visual stimulation in patients experiencing concurrent visual hallucinations. This suggests that PD patients with visual hallucinations have a decreased responsiveness to external visual stimuli [9].

Recent studies have implicated the role of the cerebellum in cognitive processes [10,11,12,13]. Cerebellar cognitive affective syndrome is characterized by executive dysfunction, spatial cognitive impairment, language deficits, and personality changes [14,15,16,17,18]. The human cerebellar cortex is a complex structure as its surface is more tightly folded than the cerebral cortex and accounts for almost 80% of the surface area of the neocortex. The nerve fiber connections to the brain’s cognitive network are extensive [19], indicating that the cerebellum plays a vital role in the development of behavior and cognition.

Many imaging studies have confirmed that cerebellar FCs are associated with cognitive networks [20,21,22], particularly with the default mode network (DMN) [23]. The posterior cerebellum (i.e., lobule VI-Crus I, lobule Crus II-VIIB, and lobule IX) is critical for cognitive representation [24,25,26]. Lobule VI, VIIB, and Crus I are explicitly involved in executive functions, including working memory, planning, organizing, and strategy formation. These functions are all important for divergent creative thinking [27,28,29,30]. Yao et al. found that the left lobule VI, VIIB, Crus I, and Crus II of the cerebellum are significantly associated with visual activity and visual divergent thinking, such as photography and drawing [31]. Yao et al. conducted a study using a combination of VBM and FC resting-state fMRI to investigate the presence of hallucinations in patients. The study found no significant reduction in gray matter volume but did observe a significant increase in DMN activation compared to patients without hallucinations [10]. When comparing hallucinators to controls in both the Charles Bonnet Syndrome and Parkinson’s disease groups, lower gray matter volume was observed in cerebellar lobule VIIIb, VIIIa, IX, and VIIb, predominantly within PD [32].

To investigate whether the cerebellum regulates the production of visual hallucinations, we conducted a study and found that the cerebellar vermis plays an important role in PD-H. Objective manifestations of cortico-cerebellar FC occur in PD patients, and the characteristics and differences of cerebellar FC in PD-H, PD-NH, and HC remain. This study validated the effectiveness of cortico-cerebellar features in a classification task across all subjects using SVMs. The sensitivity and specificity of the classifiers were also evaluated for various tasks, and the confidence intervals for each result were calculated.

## 2. Materials and Methods

### 2.1. Participants

All study subjects were recruited from Nanjing Brain Hospital between November 2018 and October 2022. All subjects were right-handed, and the study included 17 PD-H subjects (7 males and 10 females), 13 PD-NH subjects (5 males and 8 females), and 26 HC subjects who were not statistically different in terms of age and sex (12 males and 14 females). All participants provided written informed consent, and the study was approved by the Medical Research Ethical Committee of Nanjing Brain Hospital in Nanjing, China.

The diagnosis of Parkinson’s disease adheres to the criteria outlined in the “Diagnostic Standards for Parkinson’s Disease in China (2016 Edition)”. Additionally, a follow-up period of over 1 year was conducted to exclude patients with notable cognitive decline and those with concurrent Parkinson’s syndrome, such as Lewy body dementia. All participants underwent comprehensive and standardized neuropsychological assessments to evaluate their cognitive function, including the Montreal Cognitive Assessment (MoCA) [33], the Mini-Mental State Examination (MMSE) [34], and the Unified Parkinson’s Disease Rating Scale Part 1 (UPDRS-1).

The inclusion criteria for HCs were as follows: (1) absence of current cognitive problems, (2) absence of neurological or psychiatric disorders, and (3) a clinical dementia score of 0.

The general information group analysis of the three participant groups is shown in Table 1. The chi-square test was used for gender, while an ANOVA (analysis of variance) was used for the rest. P1, P2, and P3 were the results of multiple comparisons between Group 1 (HC) and Group 2 (PD-H), Group 1 and Group 3 (PD-NH), and Group 2 and Group 3, respectively. A significance level of *p* < 0.05 was used to determine statistical significance. From the table, it can be observed that there are no significant differences in age, education level, and gender among the three groups. Additionally, there is no significant difference in the course of disease between the PD hallucination group and the PD-NH group.

From this table, it can be seen that the cognitive situation of healthy elderly individuals differs from that of PD patients. Considering that previous studies have shown that patients with PD experience symptoms of cognitive decline, a further comparison of the cognitive status between the PD hallucination group and the PD-NH group revealed no significant difference between the two groups. Conclusion: there is no statistically significant difference in the general information of these data, which can be utilized for further research.

### 2.2. Structure Magnetic Resonance Imaging Data Acquisition and Preprocessing

Magnetic resonance imaging was acquired using a Siemens 3.0 T single scanner (Siemens, Verio, Germany) with an 8-channel radio frequency coil at the Affiliated Brain Hospital of Nanjing Medical University. Participants were instructed to maintain stillness, close their eyes, stay awake, and refrain from thinking about anything. T1-weighted imaging (T1WI) was acquired using a three-dimensional magnetization-prepared rapid gradient echo (3D-MPRAGE) sequence. The parameters included time repetition (TR) = 2530 ms, echo time (TE) = 2.48 ms, inversion time (TI) = 1100 ms, number of slices = 176, thickness = 1 mm, gap = 0.5 mm, matrix = 256 × 256, flip angle (FA) = 90°, field of view (FOV) = 256 × 256 mm, and voxel size = 1 mm × 1 mm × 1 mm. The imaging for each subject took approximately 8 min.

Among the subjects, there were four cases of cerebral infarction and one case of white matter lesion in each of the PD-H patients. Among the PD-NH patients, there were 4 cases of cerebral infarction, with one case complicated by old lesions in the brainstem. The other subjects showed no significant abnormalities, except for patients with large-scale cerebral infarction and brain tumors in the imaging. Figure 1 shows the structural images of the original data for the subjects in each group.

### 2.3. Functional Magnetic Resonance Imaging Data Acquisition and Preprocessing

Resting-state fMRI acquisition was performed using single echo planar imaging (EPI). The gradient echo-echo planar imaging (GRE-EPI) sequence included 240 time points. TE = 30 ms; TR = 2000 ms; number of slices = 31; FOV = 220 mm × 220 mm; matrix = 64 × 64; FA = 90°; thickness = 3.5 mm, gap = 0.6 mm. The imaging for each subject took approximately 8 min. Those in need of scientific research can contact the author, and the data can be shared with the patient’s consent.

The fMRI data were processed using Data Processing and Analysis for Brain Imaging (DPABI, http://www.restfmri.net (accessed on 1 January 2020)) [35]. The first 10 volumes of the rest session were discarded for each subject. The remaining images were corrected using slice timing and motion correction techniques (head motion ≤ 3 mm, head motion angle ≤ 3°). Next, resting-state fMRI images were co-registered with high-resolution 3D-T1 structural images. The normalization of 3D-T1 structural MRI images to the Montreal Neurological Institute (MNI) space was performed using non-linear warping based on Diffeomorphic Anatomical Registration Through Exponentiated Lie Algebra (DARTEL). After spatial normalization to T1 space, all images were resampled into 3 mm × 3 mm × 3 mm voxels and spatially smoothed using a Gaussian filter with a full-width at half-maximum (FWHM) of 6 mm. Data were then temporally bandpass-filtered (0.01–0.08 Hz) to eliminate low-frequency drifts and physiological high-frequency noise.

Furthermore, in order to minimize the influence of confounding factors such as head movements during rest and physiological noise (such as respiration and cardiac fluctuations), nuisance covariates were included in the regression analysis. These covariates consisted of the Friston 24-motion parameter model, global mean signal, white matter signal, and cerebrospinal fluid signal.

We selected three groups of subjects, and we display their fMRI thermograms after applying noise reduction in Figure 2. As depicted in the figure, there were significant differences in the distribution of thermal values between the PD-H group and the other two groups, particularly in the prefrontal region. However, in actual clinical situations, early Parkinson’s patients do not exhibit uniform and obvious abnormalities in a specific brain area. Therefore, we introduced FC with time series to conduct further research on the three groups of subjects.

We selected three groups of subjects, and we display their denoised fMRI images in Figure 2, as shown in the figure. By comparing only the images, early Parkinson’s patients did not exhibit any obvious lesion features on their functional images, regardless of whether they experienced hallucinations or not. At a certain moment, there were no apparent abnormalities in any brain region. Therefore, we introduced FC with time series to further study the three groups of subjects.

### 2.4. Functional Connectivity Analysis

From a structural anatomy standpoint, the bilateral cerebellum (26 regions) can be divided into the cerebellar hemisphere (18 regions) and vermis (8 regions), as shown in Figure 3. These regions were extracted as the seed regions of interest using the DPABI software package template, specifically the anatomical automatic labeling (AAL) template. An FC analysis was performed between each seed region and the whole brain (115 regions) using the DPABI software in a voxel-wise manner. The voxels of each seed region from every subject were extracted and averaged to obtain the reference time series of seed points. Then, we calculated the correlation coefficient between the reference time series and the time series of all other brain voxels. This part obtained 2990 connectivity features (115 × 26 = 2990).

### 2.5. Feature Select 

The correlation coefficients were transformed into z-values using the Fisher r-to-z transformation to improve normality and enhance the classification performance of machine learning. All data were tested for normality and variance homogeneity. One-way ANOVA tests were performed to compare the FC values across groups (PD-H, PD-NH, and HC).

We then used post hoc multiple comparisons to compare the results between pairs of groups. We considered *p* < 0.05 as statistically significant in our dataset, which was used as a training feature in a classification task. A finding with a *p*-value < 0.001 is considered statistically significant and will be discussed in this article.

Three features were selected for training separately: Feature 1, all cortico-cerebellar FC; Feature 2, all ROI seed sequences that showed significance in the statistical analysis; and Feature 3, cortico-cerebellar FC that showed significance in the analysis. In the classification task, since only Feature 1 was consistent across the three groups of subjects, three sets of binary classification tasks were created: PD-H versus (vs). PD-NH, PD-H vs. HC, and PD-NH vs. HC.

### 2.6. SVM Classification 

The SVM was implemented in MATLAB (The MathWorks, Natick, MA, USA) and LIBSVM (http://www.csie.ntu.edu.tw/cjlin/libsvm/ (accessed on 11 October 2023)). The kernel function in the SVM classifier uses the radial basis kernel function (RBF). The penalty parameter, C, and the kernel bandwidth, s, in the kernel function range from [4^−4^,4^4^]. The RBF kernel was defined as follows:(1)$$K\left(x_{1},x_{2}\right)=\exp\left(-\|x_{1}-x_{2}\|/\left(2\sigma^{2}\right)\right)$$
where $x_{1}$, $x_{2}$ are two eigenvectors, and σ is the width parameter of the REF kernel. 

A 4-fold cross-validation approach was used to assess the models by training four models for each experiment. In each iteration, 75% of the data were randomly selected for training and 25% for testing. Thus, this process results in one prediction per sample from the entire dataset. This process was repeated four times to assess the variability of the evaluation metrics.

In the cross-validation, the optimal model’s result in the testing set is recorded as the classification accuracy for this task. The test set consists of two parts of data: one is the training data of the optimal model combined with white noise at a level of 10^−3^, and the other is the test data used during the training of the optimal model. The metrics of model performance included accuracy (percentage correctly classified) and the area under the receiver operating characteristics (ROC-AUC) curve.

Finally, the held-out sample was used to evaluate the training classifier. These parameters were defined as follows:(2)$$\mathrm{Accurary}\left(\mathrm{ACC}\right)=\left(TP+TN\right)/\left(TP+TN+FP+FN\right)$$
where *TP* is true positive, *TN* is true negative, *FP* is false positive, and *FN* is false negative, respectively. Area Under Curve (AUC) was defined as the area under the ROC curve and the coordinate axis.
(3)$$\mathrm{Sensitivity}\left(\mathrm{SEN}\right)=TP/\left(TP+FN\right),\;\;\;\;\mathrm{Specificity}\left(\mathrm{SPE}\right)=TP/\left(TP+FN\right)$$

The 95% confidence intervals (CIs) of ACC, SEN, and SPE were calculated using the exact Clopper–Pearson method.

## 3. Results

### 3.1. Functional Connectivity Changes of the Cerebellar

The results of the FC statistical analysis with a *p*-value < 0.001 for the three populations are recorded in Table 2. The significantly different FC connections in each of the two groups of subjects are presented in the form of x vs. y. Node-a represents the cerebellar ROI as the seed, and Node-b represents the node connected to Node-a. The brain regions in the table are labeled with the names from the AAL template. The abnormal FC connections are shown in Figure 4.

The FC connections with a *p*-value < 0.05 are shown in Appendix A. Although they are not discussed in this article, interested readers can refer to it.

#### 3.1.1. Functional Connectivity Changes of the Vermis Lobule I_II

In comparison to the PD-NH group, the FC value between vermis lobule I_II and the frontal lobe/bilateral IFGoprec, bilateral IFCtriang, left parietal lobe/PreCG, PoCG, and IPL were significantly increased in the PD-H group.

Furthermore, the FC values in vermis lobule I_II and the left frontal lobe/IFGoprec, IFCtriang, and left side of cerebellum lobule IV were significantly increased in the HC group compared to the PD-NH group.

#### 3.1.2. Functional Connectivity Changes of the Vermis Lobule IV_V

In comparison to the HC group, the FC value between vermis lobule IV_V and the left parietal lobe/PreCG was significantly increased in the PD-H group. There were no significant differences in the FC values of vermis lobule IV_V and the whole brain between the other groups.

### 3.2. Machine Learning Training Results

The training results of the cerebellar features for the three populations are shown in Table 3. The training results of Feature 1, Feature 2, and Feature 3 are recorded, respectively.

The average result of the k-fold cross-validation for each group was recorded along with the area under the ROC curve (AUC) of the corresponding models. The CI for each result value is recorded in square brackets, [], next to that value.

In the classification tasks for the PD-H and HC groups, the Feature 1 model has the highest accuracy, and it is more sensitive to the HC group. In the classification task for the PD-H and PD-NH groups, the Feature 2 model has the highest accuracy and is optimal in all parameters.

In the classification task for the PD-NH and HC groups, the model using Feature 1 achieves the highest accuracy and AUC and is more sensitive to the HC group. On the other hand, Feature 2 and Feature 3 are more sensitive to the PD-NH group.

Figure 5 shows the ROC curves of the Feature 1, Feature 2, and Feature 3 training models for the three sets of tasks. All FC connections in the cerebellum (Feature 1) have achieved the best performance, suggesting that in addition to the significant brain regions mentioned earlier, other regions in the cerebellum can also serve as biomarkers. These regions have the potential to classify patients as having or not having hallucinations. This is also reflected in Feature 2, indicating that all brain regions in the cerebellum may be more or less involved in the regulation of hallucinations.

## 4. Discussion and Conclusions

In the present study, we investigated alterations in FC between the cerebellum and the whole brain among subjects with PD-H and PD-NH and HCs. We designed three groups of cerebellar features for machine learning training in order to classify three groups of subjects, and we achieved good classification results. We also explored the relevance of this change to visual hallucinations. 

The current research indicates that there are significant changes in the FC of cerebellar cognitive subregions within the PD-H groups. The research on the brain functional connections of PD-H is now a hot topic [36], but few studies have evaluated the effect of cerebellar functional connections.

### 4.1. Functional Connectivity Changes in the PD-H Patients

In the present study, the changed FCs in the PD-H group, as shown in Table 2, were primarily observed in the ventrolateral prefrontal cortex (VLPFC). These changes included the right side of IFGoprec and IFCtriang and the DMN, including the left side of the postcentral gyrus, parietal lobe, and precuneus.

The DMN is a brain resting network that becomes active when individuals are not engaged in attending to or responding to external stimuli. It is involved in regulating self-reflection and memory processes [37,38,39], which can sometimes lead to hallucinations. 

Previous studies have shown that interruptions in the DMN and its internal functional connections are the basis for mild hallucinations in PD [8]. Furthermore, a strong relationship between DMN, lobule IX, and crus II has been reported previously [40]. In this study, it was found that, compared to the PD-NH group, the FC connection between the cerebellar vermis and the DMN was also significantly increased in the PD-H group. Thus, the increased connectivity of cerebellar function to the DMN in PD-H patients may be one of the causes contributing to visual hallucinations.

In the comparison of the PD-H and HC groups, only the connection between Vermis_4_5 and the left side of PCUN was found to be significant in the FC with *p* < 0.001 in Table 2. This finding also emerged in the group comparison between PD-H and PD-NH, indicating that the left side of the precuneus is a common brain area that frequently interacts with the cerebellum in PD-H patients.

The precuneus is a component of the posterior parietal cortex, situated in the inner hemisphere of the brain. It plays a crucial role in the DMN. The cognitive function of the brain involves situational memory, visual spatial processing, self-related information processing, as well as metacognition, consciousness, and other processes. It is a complex cognitive ability for individuals to assess the accuracy of their memory.

Some research has also shown that a more active precuneus allows for increased information flow in the brain, facilitating connections between unrelated things. When schizophrenia patients perform memory tasks, they exhibit reduced inhibition of the precuneus. Therefore, the generation of visual hallucinations may also be a result of memory dysfunction, in which the cerebellum also plays a role in regulating them.

For the enhanced connectivity between Vermis_1_2 and the VLPFC in the PD-H group, according to the domain-specific hypothesis [41], the VLPFC primarily controls visuospatial and shape information. It transfers this information to other brain regions through parallel loops, enabling the representation of both types of information in the dorsolateral and ventral lateral prefrontal cortex. In the ^18^F-FDG-positron emission tomography study [42], PD patients with non-dementia type and visual hallucinations (VH) showed a significant increase in local brain glucose metabolism rate in the frontal lobe, as well as a decrease in glucose metabolism rate in the temporal parietal occipital lobe. This also provides evidence of abnormal activity in the frontal lobe of PD-H patients, which may be regulated by the cerebellum in this study.

The over-activation of connectivity in the PD-H group leads to an imbalance of visual information across brain regions, resulting in hallucinations. This process also involves the cerebellar vermis.

### 4.2. Functional Connectivity Changes in the PD-NH Patients

The main cerebellar abnormality observed in the PD-NH group subjects, compared to the other two groups, was in Vermis_1_2, which showed a significant reduction.

In the comparison of the PD-NH and HC groups, there was a significant connection between Vermis_1_2 and the left side of IFGoprec, IFCtriang, and cerebellar lobule IV_V in FC (*p* < 0.01, Table 2). These brain regions are primarily responsible for motor functions, and the weakened connections between the cerebellum and its functional connectivity may be associated with motor deficits in PD patients.

### 4.3. Functional Connectivity Changes in Classification Task

In the classification task of PD-H versus HC, the model ACC gradually decreased as the number of cerebellar FC connections decreased. This suggests that multiple groups of cerebellar FC connections exhibit abnormality and classification sensitivity in PD-H patients compared to HCs. The model was more sensitive to the HC subjects, with a screening rate of 96.15%, while it only reached 88.24% for the PD-H subjects. The model accuracy decreased significantly during the training of Feature 3, likely due to the low number of FC connections with significance.

In the classification task of PD-H versus PD-NH, the model achieved optimal results in terms of accuracy (ACC) for Feature 2. This suggests that the cerebellar vermis in the two subject groups exhibits distinct patterns of brain modulation. Furthermore, including functional connectivity (FC) connections from other parts of the cerebellum reduces the classification sensitivity of this particular feature. The model is more sensitive to PD-H subjects, with a screening rate of 94.12%.

In the classification task of PD-NH with HCs, the model ACC obtained the best results in Feature 1. The classification effect of the models trained with Feautre 2 and Feautre 3 was identical. However, the model AUC was still higher for Feautre 2, which means that the additional features in Feauture 2 compared to Feauture 3 still had an enhancing effect.

The cerebellar FC connections performed best in the classification task of Feature 1, indicating that most cerebellar FC connections were altered in both PD-H and PD-NH patients. However, some of these changes may not be statistically significant at the moment, but they can still be identified and utilized by the SVM model.

## 5. Conclusions

In previous studies on Parkinson’s hallucinations, the mechanism behind hallucination production has often been centered on the brain. The cerebellum vermis has been traditionally associated with motor regulation. However, based on experimental findings, it seems to be closely connected to the DMN and plays a significant role in regulating cognition related to hallucinations. This suggests that the cerebellum vermis may be a crucial brain region that has been overlooked in previous studies.

## Figures and Tables

**Figure 1 brainsci-13-01458-f001:**
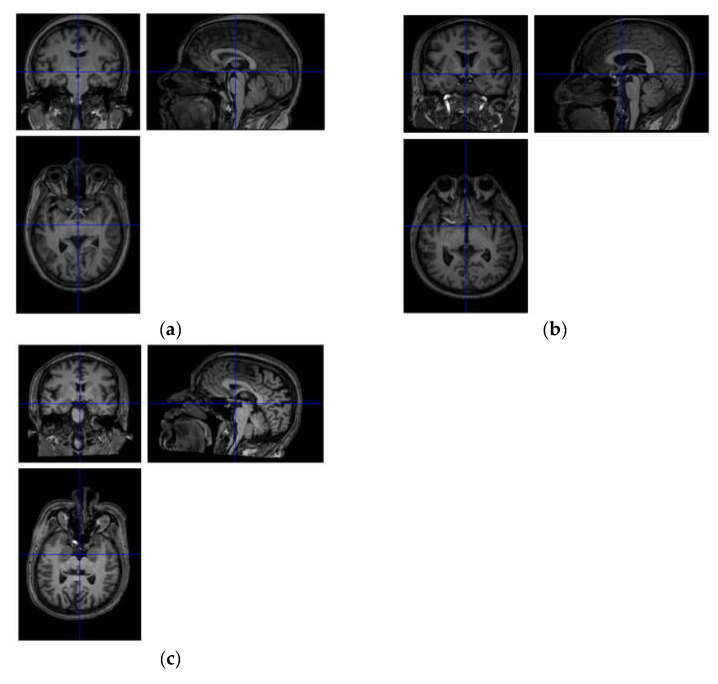
Original T1 figures (**a**) HC; (**b**) PD-H; (**c**) PD-VH.

**Figure 2 brainsci-13-01458-f002:**
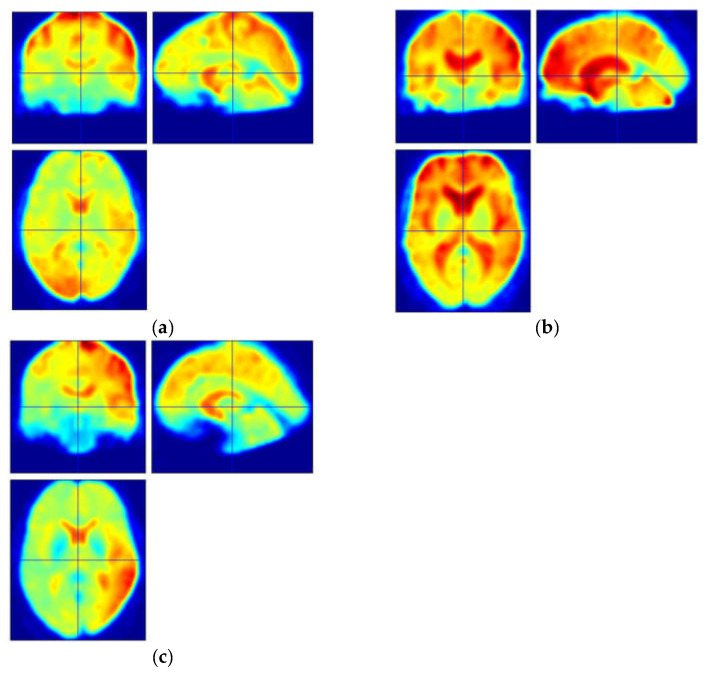
Preprocessed original fMRI images (**a**) HC; (**b**) PD-H; (**c**) PD-VH.

**Figure 3 brainsci-13-01458-f003:**
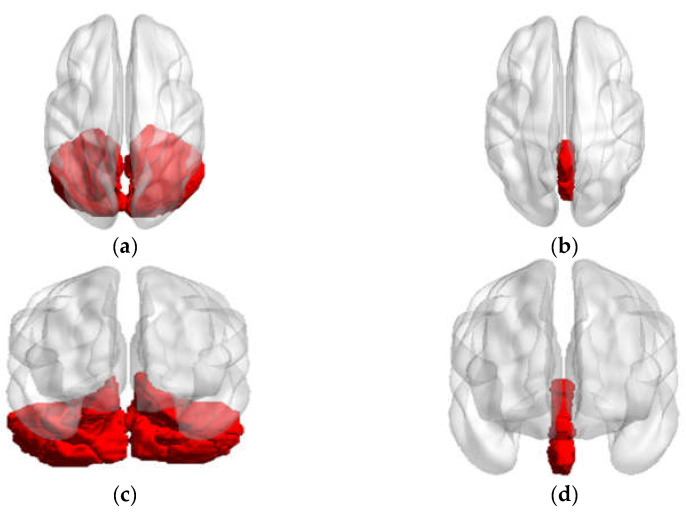
The axial and conronal views of seed regions. (**a**) Axial views of cerebellum; (**b**) axial views of vermis; (**c**) coronal views of cerebellum; (**d**) coronal views of vermis.

**Figure 4 brainsci-13-01458-f004:**
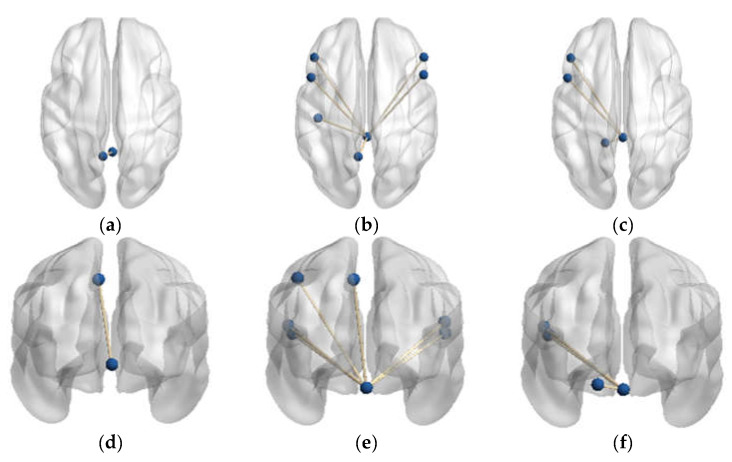
FC connections with significant differences. (**a**) Axial sight of FC between the PD-H and HC groups; (**b**) axial sight of FC between the PD-H and PD-NH groups; (**c**) axial sight of FC between the PD-NH and HC groups; (**d**) coronal sight of FC between the PD-H and HC groups; (**e**) coronal sight of FC between the PD-H and PD-NH groups; (**f**) coronal sight of FC between the PD-NH and HC groups.

**Figure 5 brainsci-13-01458-f005:**
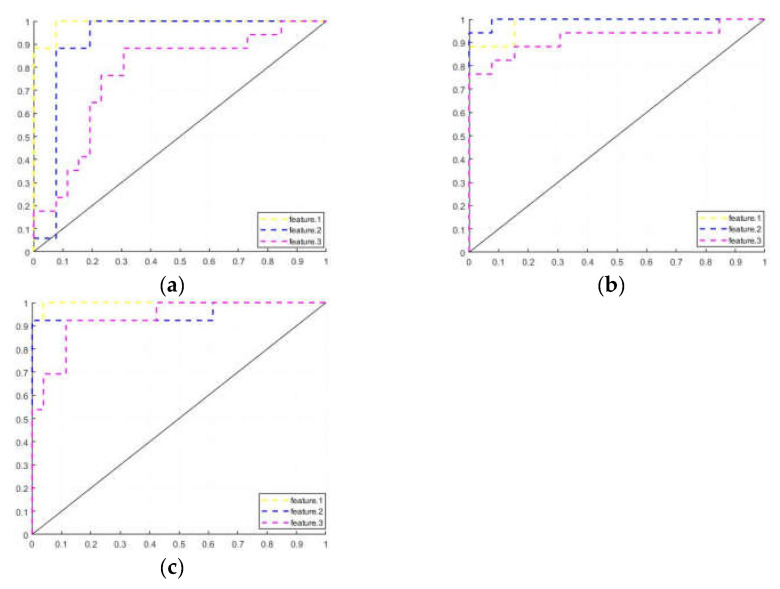
ROC curves (**a**) Feature 1; (**b**) Feature 2; (**c**) Feature 3.

**Table 1 brainsci-13-01458-t001:** Comparison of general information among three groups.

	HC	PD-H	PD-NH	*p*	P1	P2	P3
Age (years)	59.12 ± 6.289	63.24 ± 6.582	59.00 ± 6.338	0.091			
Education (years)	8.788 ± 3.287	8.294 ± 4.832	7.923 ± 4.609	0.814			
Sex (male)	46.2%	41.2%	38.5	0.285			
Moca (scores)	25.38 ± 3.601	19.76 ± 4.191	21.85 ± 5.398	0.000	0.017	0.000	0.599
Course (years)		2.294 ± 1.105	1.577 ± 1.718		-	-	0.176

**Table 2 brainsci-13-01458-t002:** FC statistical analysis results for *p* < 0.001.

PD-H vs. HC		PD-H vs. PD-NH	
Node-a	Node-b	Node-a	Node-b
Vermis_4_5	Precuneus_L	Vermis_1_2	Frontal_Inf_Oper_L Frontal_Inf_Oper_R Frontal_Inf_Tri_L Frontal_Inf_Tri_R Postcentral_L Parietal_Inf_L Precuneus_L
PD-NH vs. HC			
Vermis_1_2	Frontal_Inf_Oper_L Frontal_Inf_Tri_L Cerebelum_4_5_L		

**Table 3 brainsci-13-01458-t003:** Feature training results.

	ACC	SEN	SPE	AUC
Feature 1				
PD-H:HC	93.02 [80.94,98.54]	88.24 [63.56,98.54]	96.15 [80.36,99.90]	0.9910
PD-H:PD-NH	86.67 [69.28,96.24]	88.24 [63.56,98.54]	84.62 [54.55,98.08]	0.9819
PD-NH:HC	92.31 [79.13,98.38]	84.62 [54.55,98.08]	96.15 [80.36,99.90]	0.9970
Ferutre 2				
PD-H:HC	88.37 [74.92,96.11]	82.35 [56.57,92.30]	92.31 [74.87,99.05]	0.9140
PD-H:PD-NH	93.33 [77.93,99.18]	94.12 [71.31,99.85]	92.31 [63.97,99.81]	0.9955
PD-NH:HC	89.74 [75.78,97.13]	92.31 [63.97,99.81]	88.46 [69.85,97.55]	0.9527
Feature 3				
PD-H:HC	74.42 [58.83,86.48]	70.59 [44.04,84.69]	76.92 [56.35,91.03]	0.7715
PD-H:PD-NH	80.00 [61.43,92.29]	88.24 [63.56,98.54]	69.23 [38.57,90.91]	0.9186
PD-NH:HC	89.47 [75.78,97.13]	92.31 [63.97,99.81]	88.46 [69.85,97.55]	0.9349

## Data Availability

The data are available upon reasonable request.

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
