# Peer review of "Differences and Changes in Cerebellar Functional Connectivity of Parkinson’s Patients with Visual Hallucinations"

_brainsci, 2023, doi:10.3390/brainsci13101458_

Round 1

Reviewer 1 Report

The manuscript's title and abstract sound intriguing. The authors established diagnostic markers of participation in the development of cognitive problems in PD. However, detailed familiarization with the text does not allow answering the stated questions. In this regard, may be recommend for authors to pay attention to a number of points that can improve the perception of the material.

1. Immediately draws attention to the large number of abbreviations in the manuscript, which makes the text difficult to understand. I offer ideas to the authors and keep only generally accepted abbreviations (PD, fMRI, BOLD, etc.). Possibly original design. It is better to list terms from the international anatomical nomenclature in full.

2. The main problem in diagnosing PD is differential diagnosis next to neurodegenerative diseases. For example, “Lewy bodies, Parkinson's disease, cerebrovascular diseases, frontotemporal dementia, Creutzfeldt-Jakob disease, HIV-associated dementia, neurosyphilis, normal pressure hydrocephalus and dementia due to exposure to toxic-septic substances (heavy metals, alcohol, etc.) drugs ), metabolic disorders or mental disorders." As well as Alzheimer's and multiple sclerosis in the early stages. In this regard, it is recommended that the authors of the Guide justify the choice of PD taking into account other similar circumstances.

3. Lines 79 – 83. The introduction ends with statements. Why then the article? Traditionally, the introduction ends with a goal or objective. Currently, it is necessary to think through the goals of the study.

4. Lines 87 – 89. All subjects were right-handed, including 17 PD-H subjects (9 men and 8 women), 13 PD-NH subjects (5 men and 8 women), and 26 HC subjects did not differ statistically in age and gender (12 men and 14 women). Thus, there were 30 patients in the study. However, we do not see patient characteristics, age, medical history, or diagnostic criteria. The main question is: was there PD? Therefore, in the Methods chapter, it is necessary to expand information about the clinical criteria by which the diagnosis of PD was made.

5. Lines 95 – 108. The authors provide technical information on MRI registration. But there is nothing in the results chapter when applying this method. Necessary information must be added as well as the patients' original MRI figures describing changes in the brain must be submitted.

Features of fMRI registration are presented in this manuscript with more important points. It is better to isolate and describe this technique separately, as well as present original drawings of changes in the brain.

6. The need to ensure the preservation of Fig. 1 and 2. These are not original figures, but products of the software. Such figures require simultaneous presentation of original data.

7. Original digital data is presented in subchapter 3.2. Machine learning training results in table. 2 and fig. 3. Fig. 3 – poor quality, requires explanation. Table 2 shows strange numbers. In such studies, original numerical data and figures must be shown to support the hypothesis.

8. In the Discussion, abbreviations make it difficult to understand. It's better to decipher. It makes sense to add here a comparison with PET data from scientific literature, which are more valid.

Reviewer 2 Report

Line 31: you have translated SNpc into English as "the dense part" but no one does this.  We call it substantial nigra pars compacta, using the Latin term

Line 34 should say "autonomic dysfunction"

Line 39: do visual hallucinations really increase the death rate, or are they just associated with it? Likely just an association...

Lines 51-52 the English grammar is wrong; not clear what exactly is meant here; please correct and clarify

Lines 66-67 do not belong here.  This paragraph is a discussion of the normal role of the cerebellum in cognition, but these two lines describe some PD-associated pathology.

The three paragraphs Lines 59-83 are a mixed description of normal and pathological conditions.  Please reorganize all this to describe normal first and then the pathological changes

Lines 127-130 seem to be garbled or missing some text

Fig 1: should be axial and coronal "views" not "sight"

Line 151: it is Natick, not Natwick

Discussion and Conclusions: Please shorten this section and confine it to interpretation of how the results relate to your hypothesis on what FC changes occur in the two groups of PD patients.  Additionally, although the abstract discusses the usefulness of these findings and machine learning in early diagnosis, this important implication is not included in the discussion. It would also be useful to discuss whether these changes reflect anatomical alterations or re-routing of information flow in existing anatomical pathways. 

The English is understandable for the most part but there are exceptions as pointed out in some of my comments above.  Careful proof reading for complete sentences is recommended

Reviewer 3 Report

The research objectives were clearly defined from the outset. The study focused on exploring changes in the functional connectivity of the cerebellum in Parkinson's disease patients with visual hallucin

The use of resting-state functional magnetic resonance imaging (rs-fMRI) to collect data on three groups of subjects (PD-H, PD-NH and HC) is a powerful method for studying brain connectivity. In addition, the use of support vector machines (SVM) to classify objects according to cerebellar connectivity characteristics is an interesting approach.

the noteworthy concerns I have are:

1-Sample size:

The study had a relatively small number of participants, particularly in the PD-H group.

 A larger sample size may increase the reliability of the results.

“To strengthen the validity of the results, it would be useful if other researchers could reproduce the study with independent samples.”

2- This study provides interesting data on the differences in cerebellar connectivity between the groups, but it is important to discuss the clinical implications of these results in more detail. How might these differences in connectivity influence the perception and occurrence of visual hallucinations in patients with Parkinson's disease?

3- It is preferable to revise the presentation of results to make them clearer

In conclusion, this article makes a significant contribution to our understanding of the neural mechanisms underlying visual hallucinations in patients with Parkinson's disease. These findings may have important implications for the diagnosis and management of this disease.

Reviewer 4 Report

In the article entitled "Differences Changes in Cerebellar Functional Connectivity of Parkinson's patients with visual hallucinations," the authors investigated impaired functional connections in Parkinson's disease patients with hallucinations and their impact on cognitive performance. The authors demonstrate changes in cerebellar functional connectivity in patients with hallucinations.

Some observations.

1.         The sample size is relatively small, which may limit the generalizability of the results.

2.         The study was only focused on functional connectivity in the cerebellum and its cognitive subregions in Parkinson's patients presenting with hallucinations.

3.         Other brain regions and their functional connectivity patterns, which could provide additional information on the neural mechanisms underlying hallucinations, were not explored.

4.         The study used resting-state functional MRI data, which only captures spontaneous brain activity. However, this approach does not provide information on dynamic changes in functional activity during cognitive tasks or processes.

5.         The study did not investigate longitudinal changes in cerebellar functional connectivity in Parkinson's patients with hallucinations, which could provide important information on the progression of hallucinations.

6.         It is not specified if Parkinson's patients were taking medication, as it has been shown that there is increased cerebellar connectivity in OFF medication patients and decreased connectivity in ON medication patients.

The results of the research could be a good pilot study that gives a guideline for further studies.

Round 2

Reviewer 1 Report

1. Pictures in the appendix? not described. Pictures can be placed in the body of the manuscript with a detailed description and correct title.

2. The manuscript deals with the analysis of fMRI data. In the first review, I already said that it is necessary to provide the original fMRI data with a description. However, I don't see this information.
